# Anti-Obesity Effects of Multi-Strain Probiotics in Mice with High-Carbohydrate Diet-Induced Obesity and the Underlying Molecular Mechanisms

**DOI:** 10.3390/nu14235173

**Published:** 2022-12-05

**Authors:** Hye Rim Kim, Eunsol Seo, Seyeon Oh, MinYeong Seo, Kyunghee Byun, Byung-Yong Kim

**Affiliations:** 1CH Labs Corp., Seoul 07249, Republic of Korea; 2Functional Cellular Networks Laboratory, Lee Gil Ya Cancer and Diabetes Institute, Gachon University College of Medicine, Incheon 21999, Republic of Korea; 3Shinwoo Co., Ltd., Changwon 14067, Republic of Korea; 4Department of Anatomy & Cell Biology, Gachon University College of Medicine, Incheon 21936, Republic of Korea

**Keywords:** probiotics, obesity, high-carbohydrate diet, Firmicutes/Bacteroidetes ratio

## Abstract

Overconsumption of highly refined carbohydrates contributes significantly to the current obesity pandemics. Probiotic administration protects against weight gain in animals fed a high-fat diet (HFD). Nonetheless, the anti-obesity effects of probiotics in a high-carbohydrate diet (HCD)-induced obesity models are not well elucidated. Herein, C57BL/6N male mice were fed an HCD (70% kcal carbohydrate) for 12 weeks and were orally treated with multi-strain probiotics (MSPs) at 10^10^ CFU or saline every day for 6 weeks. MSPs contained *Lactobacillus acidophilus* DSM 24936, *Lactiplantibacillus plantarum* DSM 24937, and *Limosilactobacillus reuteri* DSM 25175. MSPs treatment not only ameliorated weight gain but also modulated the body fat composition altered by HCD. The MSPs also attenuated the expression of adipogenesis- and lipogenesis-related genes in HCD-fed mice. In addition, MSPs promoted the expression of lipolysis- and fatty acid oxidation-promoting factors in HCD-fed mice. Furthermore, MSPs modulated the expression of thermogenesis-related genes and the serum levels of obesity-related hormones altered by HCD. Treatment with MSPs positively reversed the Firmicutes/Bacteroidetes ratio, which is associated with a risk of obesity. Hence, this study explores the multifaceted anti-obesity mechanisms of MSPs and highlights their potential to be used as effective weight-management products.

## 1. Introduction

The number of people with obesity has reached more than 650 million worldwide [1], highlighting the compelling need to develop novel and effective strategies to prevent and manage this global pandemic. The development of obesity is a multifaceted process that is attributed to dietary, genetic, and environmental factors [2]. People are prone to gain weight when these factors, combined with a sedentary lifestyle, result in long-term excessive calorie intake [2]. Given the westernization of dietary patterns, people are consuming increasing amounts of industrially processed foods rich in carbohydrates and fats [3]. In particular, overconsumption of highly refined carbohydrates is one of the major contributors to the development of metabolic disorders such as obesity, insulin resistance, and diabetes [3]. In fact, the WHO recommends consuming less than 5% of calories per day from added sugars [4]. The carbohydrate–insulin model suggests that the metabolic effects of carbohydrate-rich diets contribute to the current prevalence of obesity; the high consumption of refined carbohydrates increases postprandial insulin levels, which induce channeling of circulating fuel into adipose tissues rather than into non-adipose tissues [5]. Consistent with this hypothesis, Ludwig et al. showed, in a clinical trial, that a low-carbohydrate diet (20% of total energy intake) promoted the greatest energy expenditure during a weight-loss maintenance period compared to a high- and moderate-carbohydrate diet [6].

Mounting evidence supports the notion that gut microbiota is a key modulator of host metabolism and might contribute to the development of metabolic diseases [7]. Obesity (accumulation of an excessive amount of body fat) and type-2 diabetes (impairment in the regulation and usage of glucose as an energy source due to insulin resistance) are the main metabolic disorders linked with the gut microbiome. For instance, compared to conventional murine models, germ-free mice are protected from calorie-dense diet-induced obesity [7]. Therefore, rewiring the gut microbiota represents a new avenue for tackling various metabolic disorders [8]. An effective strategy to modulate the gut microbiota is the administration of probiotics, defined as “live microorganisms that, when administered in adequate amounts, confer a health benefit on the host” [7,8]. Probiotic treatment ameliorates weight gain in animals fed a high-fat diet (HFD) [9,10]. However, information on anti-obesity effects of probiotics using high-carbohydrate diet (HCD) animal models is scarce [11]. Hence, more in-depth research using models of HCD is warranted to provide a better understanding of the anti-obesity effects and molecular mechanisms of probiotics, while also reflecting western diets.

In this study, we assessed the anti-obesity effects and mechanisms of multi-strain probiotics (MSPs) containing *Lactobacillus acidophilus* DSM 24936, *Lactiplantibacillus plantarum* DSM 24937, and *Limosilactobacillus reuteri* DSM 25175 in mice with HCD-induced obesity. A previous in vitro study showed that these MSPs inhibit the activities of both α-amylase and α-glucosidase, which play critical roles in the degradation and absorption of carbohydrates in the gastrointestinal tract and influence the levels of circulating fuel (unpublished data). Our results demonstrate the potential of MSPs to be developed into effective weight-management probiotic products for people who frequently consume carbohydrate-rich foods.

## 2. Materials and Methods

### 2.1. Probiotic Strains

All *Lactobacillus* strains and the multi-strain formula used in this study were provided by SynBalance srl (Origgio, VA, Italy). The strains used were *Lactobacillus acidophilus* PBS066 (DSM 24936), *Lactiplantibacillus plantarum* PBS067 (DSM 24937), and *Limosilactobacillus reuteri* PBS072 (DSM 25175). The multi-strain formula contained equal CFU of each strain. The strains were cultured in de Man, Rogosa, and Sharpe (MRS) medium (cat. 288130; BD Co., Franklin Lakes, NJ, USA) for 24 h at 37 °C. Heat-killed strains were prepared by heating the bacterial culture in phosphate-buffered saline (PBS) (cat. 10010072; Gibco^TM^, Grand Island, NY, USA) for 15 min at 100 °C. All experiments were conducted using three independent subcultures of the strains.

### 2.2. Oil Red O Staining

The 3T3-L1 mouse preadipocyte cells were cultured in Dulbecco’s modified Eagle’s medium (cat. 10-013; Corning, Glendale AZ, USA) containing 10% (*v*/*v*) fetal bovine serum (cat 100-500; Gemini Bio-Products Inc., West Sacramento, CA, USA) and 1% (*v*/*v*) antibiotic-antimycotic (cat 15240096; Gibco^TM^) at 37 °C with 5% CO_2_. Adipocyte differentiation was performed using a differentiation kit (cat DIF001; Sigma-Aldrich, St. Louis, MO, USA) according to the manufacturer’s protocol. The heat-killed strain or orlistat was administered when the medium was replaced with maintenance medium after induction.

The differentiated cells were fixed with 4% formalin for 60 min and further treated with 0.5% (*w*/*v*) Oil Red O (cat 1320-06-5; Sigma-AldrichAfter a 60 min incubation, the stained cells were observed under a microscope (CKX53, Olympus, Tokyo, Japan) using cellSens imaging software v.2.2 Olympus). The cells were then dissolved in isopropanol and the absorbance was measured at 510 nm using a microplate reader (BioTek Instruments, Inc., Winooski, VT, USA).

### 2.3. Experimental Murine Model

C57BL/6N male mice (6 weeks old) were purchased from Orient Bio (Sungnam, Republic of Korea). Following a week of acclimation, the mice were randomized into several groups as follows: (1) the normal group was fed with a normal diet (ND) for 12 weeks and then orally administered normal saline for 6 weeks while being fed ND, (2) the HCD control group was fed a 70% carbohydrate diet (HCD; ENVIGO, Indianapolis, IN, USA) for 12 weeks and then orally administered normal saline for 6 weeks while being fed HCD, and (3) the MSP group was fed HCD for 12 weeks and then orally administered MSPs (1.0 × 10^10^ CFU/day) for 6 weeks while simultaneously fed with HCD. All mice were weighed weekly, and the blood and adipose tissues were collected on the last day of the 6th week of the oral administration period. The experiment was approved by the ethical principles of the IACUC of Gachon University (approval number: LCDI-2021-0060).

### 2.4. Sample Preparation

#### 2.4.1. RNA Extraction

The homogenized fat tissues were stored for 5 min at room temperature and centrifuged at 12,000× *g* and 4 °C for 5 min. Next, the supernatant was separated into a new tube with 0.2 mL of chloroform. The mixture was vortexed vigorously. The samples were further stored for 5 min at room temperature and centrifuged at 12,000× *g* and 4 °C for 15 min. The upper layer was carefully removed and added to a new tube containing 0.5 mL isopropanol. This mixture was stored for 10 min at room temperature and centrifuged at 12,000× *g* for 10 min at 4 °C. The solution was discarded and centrifuged at 7000× *g* and 4 °C for 5 min with 1 mL of 75% ethanol. The solution was discarded, leaving only the RNA pellet. The pellet was dried and dissolved in 50 µL RNase-free water.

#### 2.4.2. cDNA Synthesis

RNA was quantified using a Nanodrop spectrophotometer (ND-2000; Thermo Scientific, Waltham, MA, USA). RNA (1 µg) was mixed with 1 µL oligo dT primer, 1 µL dNTP mixture, and RNase-free water to achieve a final volume of 10 µL. The solution was incubated for 5 min at 65 °C. Next, 4 µL of 5× PrimeScript buffer, 0.5 µL of RNase inhibitor, 1 µL of PrimeScript RTase, and 4.5 µL of RNase-free dH_2_O were added to the sample. The mixture was incubated for 45 min at 42 °C and then at 95 °C for 5 min. The prepared sample was then cooled on ice.

### 2.5. Quantitative Real-Time PCR

qRT-PCR was performed using the CFX384 TouchTM Real-Time PCR detection system (Bio-Rad Laboratories, Hercules, CA, USA). cDNA (200 ng) was added to a new tube with 5 µL of SYBR premix (Takara, Tokyo, Japan) and 0.4 µM primers (forward and reverse), and the cycle numbers were analyzed using the CFX ManagerTM software (Bio-Rad Laboratories).

### 2.6. Enzyme-Linked Immunosorbent Assay

Leptin, adiponectin, and insulin were quantified using specific ELISA Kits (cat. ab199082, ab226900, ab277390, respectively; Abcam, Cambridge, UK).

### 2.7. Microbiome Analysis

#### 2.7.1. Microbial DNA Extraction and Library Construction

Microbial gDNA was isolated from feces using the DNeasy PowerSoil Pro kit (QIAGEN, Hilden, Germany). Sequencing libraries were constructed using the Illumina 16S metagenomic sequencing library. The PCR protocol included multiple cycles: 3 min at 95 °C, 25 cycles of 30 s at 95 °C, 30 s at 55 °C, and 30 s at 72 °C, followed by a 5 min final extension at 72 °C. A universal primer pair with Illumina adapter overhang sequences was used for the first amplification (V3-F: 5′-TCGTCGGCAGCGTCAGATGTGTATAAGAGACAGCCTACGGGNGGCWGCAG-3′, V4-R: 5′-GTCTCGTGGGCTCGGAGATGTGTATAAGAGACAGGACTACHVGGGTATCTAATCC-3′). To construct the final library, the first purified 2 µL PCR product was PCR-amplified containing the index using NexteraXT Indexed Primer (Illumina, San Diego, CA, USA). The second PCR protocol was identical to the first, except 10 cycles were used. After purification, the final product was quantified through qPCR according to the qPCR Quantification Protocol Guide (KAPA Library Quantification kits for Illumina Sequencing platforms).

#### 2.7.2. Sequencing and Processing of 16S rRNA

Purified amplicons were added in equimolar proportions and paired-end sequenced (2 × 301 bp) on an Illumina MiSeq platform (Macrogen Co., Ltd., Seoul, Republic of Korea). The adapter and primer sequences of reads were trimmed using Cutadapt (ver. 3.2) and the raw data were filtered using the DADA2 package (ver. 1.18.0), R (ver. 4.0.3) with the following criteria: (i) exclusion of 250 bp of the forward sequence and 200 bp of the reverse sequence, and (ii) expected errors < 2. After excluding sample noises, amplicon sequence variants were produced using the consensus method of DADA2, followed by normalization of samples using QIIME (ver. 1.9). ASVs were designated as the organisms that showed the highest similarity (above 85% query coverage) using Blast+ (ver. 2.9.0). Multiple alignments of ASVs were performed using MAFFT (ver. 7.475) and the phylogenetic tree was produced using FastTreeMP (ver. 2.1.10). Using QIIME, alpha diversity based on the Shannon index and beta diversity based on weighted UniFrac distance were calculated.

### 2.8. Statistical Analysis

In vitro and microbiome analyses were performed using GraphPad Prism (ver. 9.3.1). The statistical method used was one-way analysis of variance (ANOVA) followed by one-way Tukey’s test for all pairwise comparisons (95% confidence interval) in in vitro analysis. The in vivo analysis was performed using *t*-tests. Differences between two groups in Firmicutes/Bacteroidetes (F/B) ratio, alpha-diversity, and relative abundance of microbiota were evaluated by Mann–Whitney test. Statistical significance was set at *p* values < 0.05.

## 3. Results

### 3.1. MSPs Suppress Lipid Accumulation In Vitro

The inhibitory activity of MSPs against lipase was comparable to that of the positive control orlistat (Figure 1A). Therefore, we treated 3T3-L1 adipocytes with MSPs to assess their effects on lipid accumulation. The MSPs group showed significantly suppressed lipid accumulation compared with the negative control group (*p* < 0.05) (Figure 1B,C). In addition, a cytotoxicity assay was performed to ensure that MSPs and orlistat did not compromise cell viability (Appendix A).

### 3.2. MSPs Ameliorate Weight Gain and Modulate Body Fat Composition in HCD-Fed Mice

Diet-induced obese mice were then orally administered with MSPs (10^10^ CFU/day) or saline for 6 weeks (Figure 2A,B). Results indicated a significant increase in body weight in the HCD group (38.30 ± 1.42 g) compared to the ND group (34.80 ± 0.92 g). Moreover, the 6-week administration of MSPs (34.40 ± 2.01 g) significantly reduced body weight compared to placebo (Figure 2B). In addition, as shown in Figure 2C, the body fat composition of the MSPs group (18.81 ± 3.01%) was a significant 36.57% lower than that of the HCD group (29.65 ± 2.44%).

### 3.3. MSPs Reduce Visceral Adipocyte Number and Attenuate Adipogenesis-Related and Lipogenesis-Related Gene Expression in HCD-Fed Mice

The results exhibited that the number of adipocytes in visceral adipose tissues in the MSPs group was significantly reduced compared to that in the HCD group (Figure 3A,B). Therefore, we speculated that the regulation of adipogenesis-related gene expression might partially mediate the inhibitory effects of MSPs on adipogenesis. The expression of adipogenesis-promoting genes, such as peroxisome proliferator-activated receptor γ (PPARγ), CCAAT/enhancer-binding protein α (C/EBPα), and sterol regulatory element-binding transcription factor 2 (SREBF2), was significantly attenuated in the MSPs treatment group compared to that in the HCD group (Figure 3C–E). As alterations in lipogenesis (a mechanism involved in triglyceride storage) are associated with obesity, lipogenesis-related gene expression was measured. The expression of lipogenesis-promoting genes (ATP-citrate lyase (ACL), acetyl-CoA carboxylase (ACC), and fatty acid synthase (FAS)) was significantly lower in the MSPs group than in the HCD group (Figure 3F–H).

### 3.4. MSPs Upregulate the Expression of Lipolysis-Related Genes and Fatty Acid Oxidation-Promoting Factors in HCD-Fed Mice

As shown in Figure 4, the administration of MSPs significantly increased the expression of lipolysis-promoting factors (acyl CoA oxidase (ACO), hormone-sensitive lipase (HSL), and adenylyl cyclase (AC)) compared to placebo administration (Figure 4A–C). Furthermore, the mRNA expression of peroxisome proliferator-activated receptor alpha (PPAR-α) and carnitine palmitoyltransferase-1A (CPT-1A), which are known to induce fatty acid oxidation, was significantly upregulated by MSPs treatment compared to saline (Figure 4D,E).

### 3.5. MSPs Promote the Expression of Thermogenesis-Related Genes in HCD-Fed Mice

HCD significantly reduced the mRNA expression of uncoupling protein 1 (UCP-1) and the peroxisome proliferator-activated receptor-γ coactivator (PGC-1α) compared to ND. In contrast, the administration of MSPs significantly increased the expression of genes encoding UCP-1 and PGC-1α compared to placebo (Figure 5A,B). In alignment with the increase in the expression of thermogenesis-related genes, the MSPs significantly increased the core temperature compared to placebo (Figure 5C).

### 3.6. MSPs Regulate the Levels of Obesity-Related Hormones in HCD-Fed Mice

Treatment with MSPs modulated the serum concentrations of hormones associated with the development of obesity. The MSPs-treated group exhibited significantly lower levels of leptin and insulin compared to the HCD-only group (Figure 6A,C). In contrast, serum adiponectin levels were significantly higher in the MSPs group than in the HCD-only group (Figure 6B) (Appendix A).

### 3.7. MSPs Modulate the Intestinal Microbiome in HCD-Fed Mice

Based on taxonomic profiling (Figure 7A), microbiota composition was altered by HCD and probiotic supplementation. At the phylum level, HCD significantly increased the F/B ratio compared to ND (*p* < 0.01). In contrast, the F/B ratio of the probiotic treatment group was comparable to that of the ND group (Figure 7B). As expected, treatment with the three *Lactobacillus* strains significantly upregulated the relative abundance of *Lactobacillaceae* compared to HCD-only treatment (*p* < 0.05) (Figure 7C). In addition, MSPs treatment significantly decreased the abundance of *Roseburia* compared to HCD-only treatment (*p* < 0.05) (Figure 7D).

## 4. Discussion

Mounting evidence suggests that the gut microbiota is an essential environmental factor that influences host metabolism and obesity development [2]. One effective strategy for modulating the gut microbiota is probiotic treatment. As our knowledge of probiotics expands, they are known to not only modulate the gut microbiome but also confer physiological and metabolic benefits to the host [12]. In fact, a great number of studies have shown that the administration of probiotics reduces body fat levels and improves obesity-related metabolic markers [13]. Shin et al. showed that oral administration of a probiotic mixture for 8 weeks ameliorated body weight and improved cholesterol levels in a high-fat diet murine model [14]. Furthermore, *L. plantarum* Ln4 administration for 5 weeks in a high-fat diet mouse model prevented weight gain and triglyceride upregulation [15]. However, most anti-obesity effects of probiotics have been investigated using HFD models [11] where fat is the most energy-dense macronutrient [16]. However, relatively fewer metabolically beneficial effects of probiotics have been observed in HCD animal models [11].

Herein, we demonstrated that oral administration of MSPs containing *Lactobacillus acidophilus* DSM 24936, *Lactiplantibacillus plantarum* DSM 24937, and *Limosilactobacillus reuteri* DSM 25175 ameliorated weight gain, conferred various metabolic effects, and modulated the gut microbiota in HCD-fed mice. Prior to the in vivo study, an in vitro study confirmed that an MSP suppresses lipase activity and lipid accumulation in adipocytes. HCD intake for 18 weeks significantly increased body weight and body fat percentage compared to ND intake. Similar to our results, various animal studies have shown that long-term intake of HCD induces significant weight gain [17,18]. However, co-administration of the MSP formulation for 6 weeks (after 12 weeks of HCD) significantly ameliorated body weight and body fat percentage gain compared to HCD-only treatment.

Histological analysis of visceral adipocytes was performed to further investigate the mechanisms underlying the anti-obesity effects of MSPs. Compared with HCD-only treatment, MSPs significantly reduced the number of visceral adipose adipocytes. In support of the histological results, probiotic formulation intake attenuated the expression of adipogenesis-promoting genes (*PPARγ*, *C/EBP α*, and *SREBP2*) and lipogenesis-promoting genes (*ACL*, *ACC*, and *FAS*) compared to HCD-only treatment. The suppression of lipogenesis-promoting genes through MSPs may ameliorate hyperglycemia, known to induce oxidative stress and, thus, insulin resistance [19]. Furthermore, treatment with MSPs significantly increased the gene expression of lipolysis-promoting factors (*ACO*, *HSL*, and *AC)* and fatty acid oxidation-inducing factors (*PPARα* and *CPT-1A*) compared to HCD-only treatment. Promoting lipolysis and fatty acid oxidation is considered an effective therapeutic strategy against obesity [20]. Consistent with our observations, a study has shown that treatment with the probiotic *Lactobacillus gasseri* for 10 weeks improved lipid metabolism in HCD-induced obese mice [18]. These results suggest that MSPs exert anti-obesity effects by modulating host lipid metabolism.

In addition to modulating lipid metabolism, thermogenic stimulation is a promising approach to combat obesity by increasing energy expenditure [21,22]. Therefore, we assessed the gene expression of UCP-1 (a non-shivering thermogenesis mediator) and PGC-1α (a mitochondrial biogenesis and adaptive thermogenesis modulator). Treatment with MSPs significantly promoted the gene expression of UCP-1 and PGC-1α compared to HCD-only treatment. Supporting the above results, the core temperature of the MSPs group was significantly higher than that of the HCD group. Similar to our study, Park et al. demonstrated that the administration of *L. amylovorus* KU4 stimulated thermogenic gene expression (UCP-1 and PGC-1α) and body temperature, thus promoting the browning of white adipocytes in HFD-induced obese mice [23].

Obesity is associated with alterations in adipocyte hormone secretion, resulting from the augmentation of adipose tissue mass [24]. This study showed that MSPs can regulate obesity-related hormone levels in HCD-induced obese mice. Leptin is a hormone involved in the modulation of appetite and metabolism, and its expression is upregulated in the adipose tissues of obese individuals [24,25]. Another obesity-related hormone is insulin, which partitions circulating fuels into adipose tissues and inhibits the uptake of fuels into non-adipose tissues, resulting in reduced energy expenditure and promotion of calorie intake in HCD [26]. Our study showed that MSP administration significantly counteracted the increase in serum levels of leptin and insulin induced by HCD. In agreement with our study, Soundharrajan et al. showed that 8-week administration of a *L. plantarum* strain significantly reduced leptin gene expression in mice with HFD-induced obesity compared to the control [27]. Adiponectin is an adipokine that plays a pivotal role in obesity. Adiponectin confers protective effects against HFD-induced lipid accumulation [28]. Our results showed that MSPs significantly promoted serum adipokine levels despite the opposite effect of HCD on adipokine secretion.

As mentioned above, the administration of probiotics is an effective method for modulating the gut microbiota. Hence, we investigated the effects of MSP administration on HCD-induced obesity in mice. Our results revealed that HCD significantly increased the F/B ratio compared to ND, and this was reversed by the administration of MSPs. In fact, various studies have shown a higher F/B ratio is positively associated with the prevalence of obesity [29,30,31]. In addition, MSPs administration significantly increased the relative abundance *of Lactobacillaceae* compared to the HCD group, suggesting the three Lactobacillus strains colonized the gastrointestinal tract. Interestingly, MSPs significantly decreased the relative abundance of *Roseburia* compared to HCD only. Conversely, several animal and human studies have shown a reduction in *Roseburia* abundance in the obese group rather than in the lean group [32,33], where *Roseburia* spp. butyrate-producers, have been implicated in the modulation of glucose homeostasis. A murine model demonstrated that wheat-derived arabinoxylans significantly restored *Roseburia* spp. abundance in HFD-induced obese mice. In contrast, Ordiz et al. demonstrated that consuming starch results in an increase and decrease in *Lactobacillus* and *Roseburia* abundance, respectively, in humans [34]. These controversial results could be explained by the genus and species variability of *Roseburia* spp. when utilizing different sources of carbohydrates that influence their gastrointestinal colonization ability [35]. However, the above results must be interpreted with caution, as the statistical methods used in this study do not consider the effects of time.

In summary, this study explored the multifaceted mechanisms of action of MSPs against HCD-induced obesity in mice. Specifically, it revealed that the anti-obesity effects of MSPs were mediated through modulation of host lipid metabolism, thermogenesis, and endocrine hormone secretion. Furthermore, the administration of MSPs restored the F/B ratio of the gut microbiota altered by HCD. The multifaceted anti-obesity molecular mechanisms of MSPs established in this murine model warrant human studies to assess the potential of utilizing MSPs as an effective weight management strategy for obese individuals who frequently consume carbohydrate-rich foods.

## Figures and Tables

**Figure 1 nutrients-14-05173-f001:**
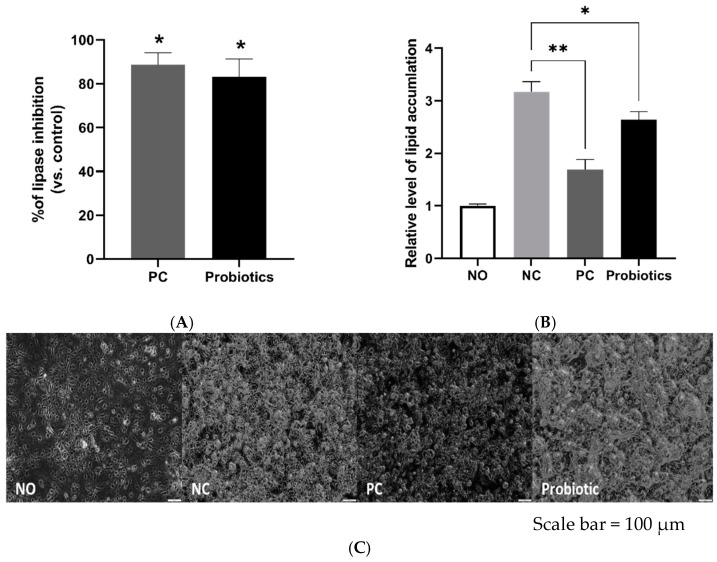
Inhibitory effect of multi-strain probiotics (MSPs) on lipid accumulation in vitro. (**A**) Relative inhibition of lipase activity (%). (**B**) Relative level of lipid content in 3T3-L1 cells. (**C**) Representative images of Oil Red O staining of 3T3-L1 cells in different groups (scale bar = 100 µm). Data are expressed as mean ± standard deviation. * *p* < 0.05; ** *p* < 0.01 vs. negative control. NO, undifferentiated 3T3-L1 cells; NC, negative control; PC, orlistat-treated; Probiotics, MSPs-treated.

**Figure 2 nutrients-14-05173-f002:**
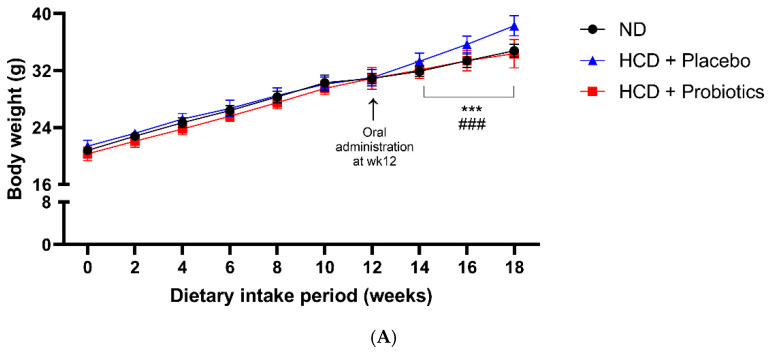
Effect of MSPs on obesity in HCD-fed mice. (**A**) Mouse body weight over the course of 18 weeks. (**B**) Effect of MSPs on body weight after 6 weeks of administration. (**C**) Effect of MSPs on body fat percentage after 6 weeks of administration. Data are expressed as mean ± standard deviation (*n* = 5 per group). *** *p* < 0.001 vs. ND; ### *p* < 0.001 vs. HCD + placebo. Legend: HCD, high-carbohydrate diet; MSPs, multi-strain probiotics; ND, normal diet.

**Figure 3 nutrients-14-05173-f003:**
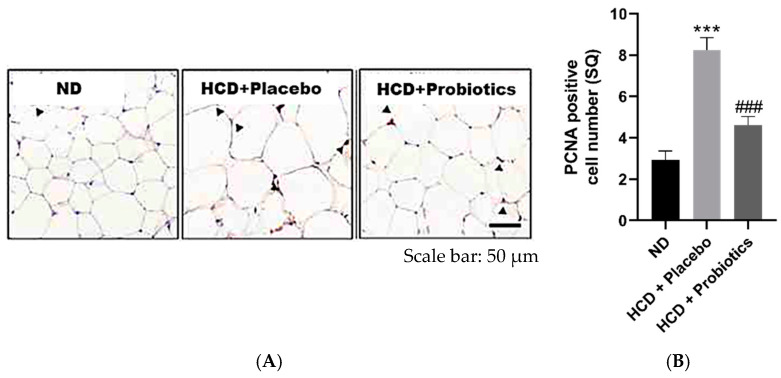
Effect of MSPs on adipogenesis and lipogenesis in the visceral adipose tissues of HCD-fed mice. (**A**) PCNA DAB staining of representative histological sections of visceral adipose tissues. Scale bar = 50 µm. (**B**) Number of PCNA-positive cells in visceral adipose tissues. (**C**–**E**) mRNA expression of adipogenesis-related genes in visceral adipose tissues. (**F**–**H**) mRNA expression of lipogenesis-related genes in visceral adipose tissues. Data are expressed as mean ± standard deviation (*n* = 5 per group). *** *p* < 0.001 vs. ND; ### *p* < 0.001 vs. HCD + placebo. Legend: *ACC*, acetyl-CoA carboxylase; *ACL*, ATP citrate lyase; *C/EBP*, CCAAT/enhancer binding proteins; *FAS*, fatty acids synthase; HCD, high-carbohydrate diet; mRNA, messenger ribonucleic acid; MSPs, multi-strain probiotics; ND, normal diet; PCNA, antibodies to the proliferating cell nuclear antigen; *PPARγ*, peroxisome proliferator-activated receptor gamma; *SREBP-1c*, sterol regulatory element-binding protein-1c.

**Figure 4 nutrients-14-05173-f004:**
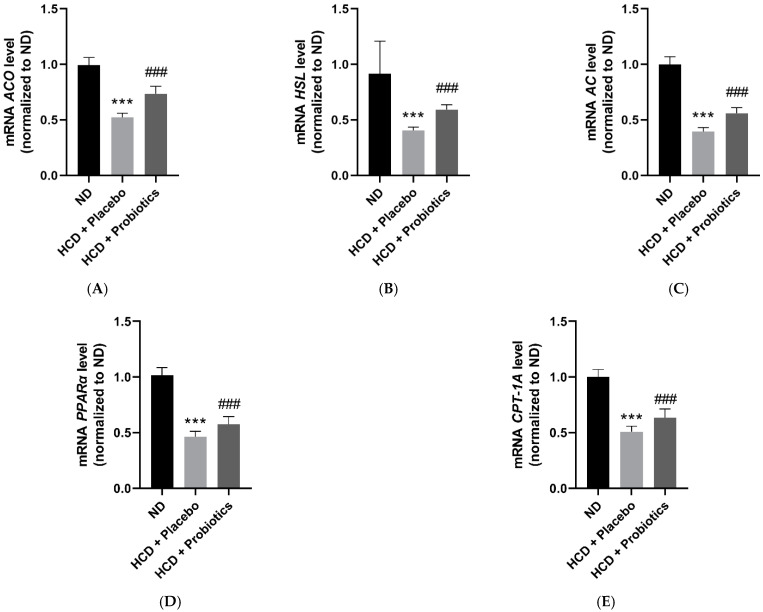
Effect of MSPs on adipolysis and fatty acid oxidation in the visceral adipose tissues of HCD-fed mice. (**A**–**C**) mRNA expression of lipolysis-promoting genes in visceral adipose tissues. (**D**,**E**) mRNA expression of fatty acid oxidation-promoting genes in visceral adipose tissues. Data are expressed as mean ± standard deviation (*n* = 5 per group). *** *p* < 0.001 vs. ND; ### *p* < 0.001 vs. HCD + placebo. Legend: *AC*, adenylyl cyclase; *ACO*, acyl CoA oxidase; *CPT-1A*, carnitine palmitoyltransferase-1A; HCD, high-carbohydrate diet; *HSL*, hormone-sensitive lipase; mRNA, messenger ribonucleic acid; MSPs, multi-strain probiotics; ND, normal diet; *PPAR-α*, peroxisome proliferator-activated receptor alpha.

**Figure 5 nutrients-14-05173-f005:**
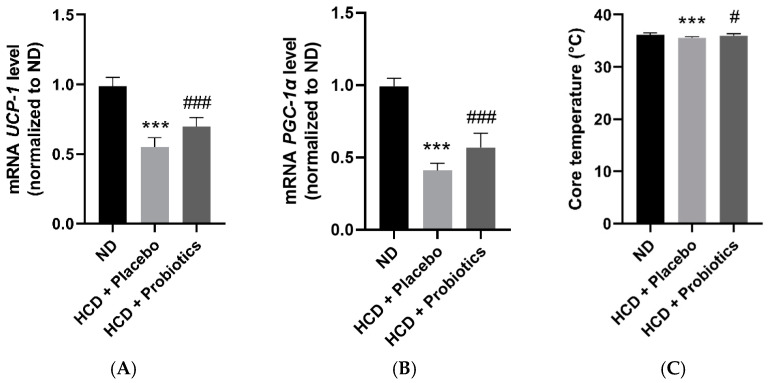
Effect of MSPs on thermogenesis in the visceral adipose tissues of HCD-fed mice. (**A**,**B**) mRNA expression of thermogenesis-related genes in visceral adipose tissues. (**C**) Core temperature of mice. Data are expressed as mean ± standard deviation (*n* = 5 per group). *** *p* < 0.001 vs. ND; # *p* < 0.05 vs. HCD + placebo; ### *p* < 0.001 vs. HCD + placebo. Legend: HCD, high-carbohydrate diet; mRNA, messenger ribonucleic acid; MSPs, multi-strain probiotics; ND, normal diet; *PGC-1α*, peroxisome proliferator-activated receptor-γ coactivator; *UCP-1*, uncoupling protein 1.

**Figure 6 nutrients-14-05173-f006:**
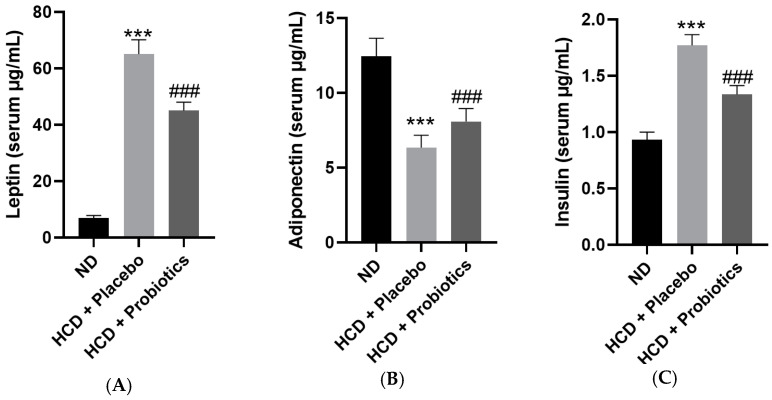
Effect of MSPs on serum adipokine levels in HCD-fed mice. (**A**) Serum leptin level. (**B**) Serum adiponectin level. (**C**) Serum insulin level. Data are expressed as mean ± standard deviation (*n* = 5 per group). *** *p* < 0.001 vs. ND; ### *p* < 0.001 vs. HCD + placebo. Legend: HCD, high-carbohydrate diet; ND, normal diet.

**Figure 7 nutrients-14-05173-f007:**
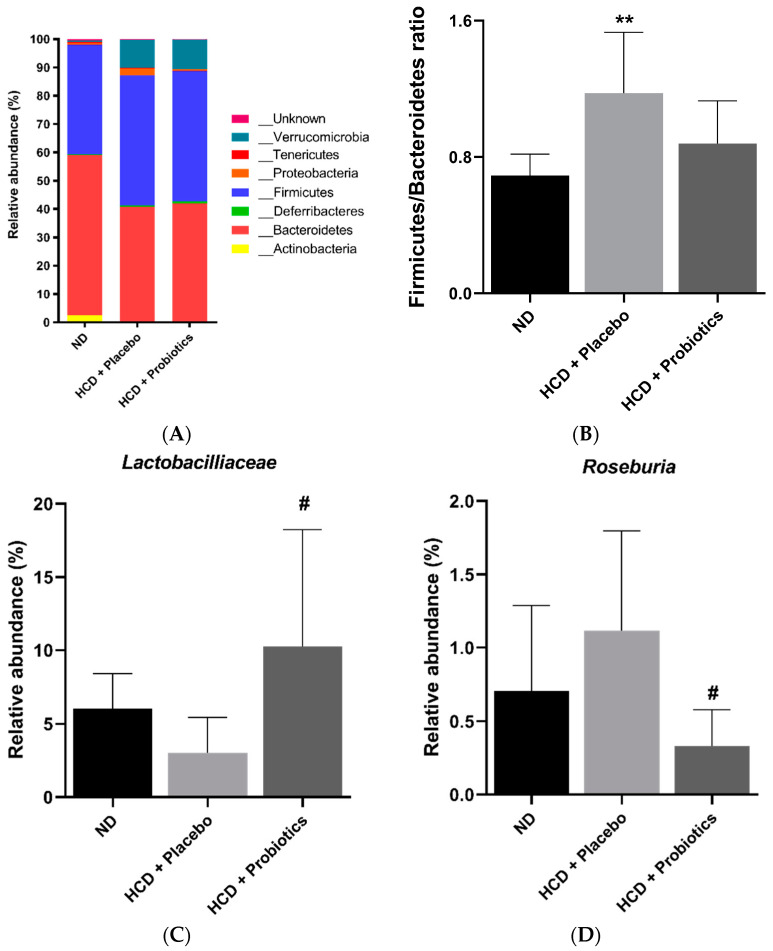
Effects of MSPs on the gut microbiota. (**A**) Bar chart of relative abundances of bacteria at the phylum level. (**B**) The Firmicutes/Bacteroidetes ratio of intestinal microbiome in mice. (**C**) Relative abundance of Lactobacilliaceae family members. (**D**) Relative abundance of the *Roseburia* genus. Data are expressed as mean ± standard deviation (*n* = 6–7 per group). ** *p* < 0.01 vs. ND, # *p* < 0.05 vs. HCD + placebo. Legend: HCD, high-carbohydrate diet; ND, normal diet.

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
