# Peer review of "Anti-Obesity Effects of Multi-Strain Probiotics in Mice with High-Carbohydrate Diet-Induced Obesity and the Underlying Molecular Mechanisms"

_nutrients, 2022, doi:10.3390/nu14235173_

Round 1

Reviewer 1 Report

Dear author my compliments for your research, may i ask you if you have any data regarding the experiments in mice ob/ob? o mice db/db? is well know the relationship of obesity and type 2 diabetes, and i believe that you may improve your introduction by adding the difference of obesity and type 2 diabetes by definition. The results are well presented but you may improve your conclusions. Can you add also a comment regarding oxidative stress obesity and type diabetes? you may take information from the paper with title "the molecular link between  oxidative stress, insulin resistance and type 2 diabetes: a targer for new therapies against cardiovasculal diseases"

Reviewer 2 Report

I suggest to the Authors to send a revised form of the manuscript that assesses the points below.

The study is carried out on animals, not immediately generalizable to humans, this should be made clear in the discussion and the title should indicate that the study was carried out on animals and to what extent the results are useful to humans.

It is important, as a good practice and for clarity, to separate the methods from the results. Only results should to be described in the results section. It is advisable to create a table, in the materials and methods section, where, for each hypothesis verified in the results (e.g.: hypothesis to test: "MSPs suppress lipid accumulation in vitro",  methods are defined: experimental ("we treated 3T3- L1 adipocytes with MSPs 177 to assess their effects on lipid accumulation...") and statistical ones ("one-way analysis of variance") .

With regard to statistical methods, it should be noted that, when we have longitudinal measures, we should use statistical methods that keep into account the effect of time in addiction to that of the treatment, such as in the analysis of variance for repeated measures. It would be advisable, for each of the hypotheses, using the repeated measures ANOVA, to compare the temporal profiles (time-treatment interaction) instead of the global averages, verifying whether the compared groups present a different temporal trend.

Finally, in my opinion, a certain conflict of interest is present, given that some Authors of this manuscript have commercial interests in the sale of nutraceuticals; this, for optimal transparency, should be indicated in the relevant section.

Reviewer 3 Report

            The article entitled "Anti-obesity effects of multi-strain probiotics in mice with high carbohydrate diet-induced obesity and the underlying molecular mechanisms" highlights the potential anti-obesity effects of probiotics. Valuable knowledge needs to be added to the experimental design as well as the discussion chapter.

1.      I recommend rephrasing the abstract. The aim of the study is unclear. The abstract should follow the style of structured abstracts: 1) Background: Place the question addressed in a broad context and highlight the purpose of the study; 2) Methods: Describe briefly the main methods or treatments applied. Include any relevant preregistration numbers and species and strains of any animals used. 3) Results: Summarize the article's main findings; and 4) Conclusion: Indicate the main conclusions or interpretations.

2.      I recommend checking the guide for citations in the text.

3.      Line 39 " In particular…." I recommend being specific by adding more information (%) "highly refined carbohydrates…".

4.      Line 45 The information from the sentence is very general. Consider rephrase-it more specifically.

5.      How many mice are in each group? Why 1010 CFU? Why 12 weeks? What factors were considered in developing the protocol?

6.      Line 100 "(1.0 × 1010 CFU/mouse)" I recommend rephrasing the information from the parenthesis because it's unclear.

7.      Results chapter- Mention the improvement in %.

8.      The discussion chapter presents too general data. The authors should focus precisely on the results they obtained and discuss them with specific original research from the domain.

9.      Line 252-269 I suggest that the paragraph is more suitable for the introduction section.

10.  Line 338-339 The microorganism name should be in italic.

11.  Line 345-346 "Nevertheless" I recommend rephrasing the sentences because it's unclear.

12.  The conclusion section is missing.

13.  References- Please check the format of the references. I recommend checking the guide for reference format.

Round 2

Reviewer 2 Report

 While nearly all of the issues raised have been addressed, questions remain about the statistical methods. The authors, in my opinion, did not adequately answer the question about statistical methods. The authors should at least comment in the discussion that the statistical methods are rather simplistic and that the effect of time has not been considered, so that the validity of the results may be quite limited.
